# TFAP2 transcription factors are regulators of lipid droplet biogenesis

Cameron C Scott[1], Stefania Vossio[1], Jacques Rougemont[2], Jean Gruenberg[1,2]*

[1]Department of Biochemistry, University of Geneva, Geneva, Switzerland;
[2]Department of Theoretical Physics, University of Geneva, Geneva, Switzerland

**Abstract** How trafficking pathways and organelle abundance adapt in response to metabolic and physiological changes is still mysterious, although a few transcriptional regulators of organellar biogenesis have been identified in recent years. We previously found that the Wnt signaling directly controls lipid droplet formation, linking the cell storage capacity to the established functions of Wnt in development and differentiation. In the present paper, we report that Wnt-induced lipid droplet biogenesis does not depend on the canonical TCF/LEF transcription factors. Instead, we find that TFAP2 family members mediate the pro-lipid droplet signal induced by Wnt3a, leading to the notion that the TFAP2 transcription factor may function as a 'master' regulator of lipid droplet biogenesis.

DOI: https://doi.org/10.7554/eLife.36330.001

*For correspondence:
jean.gruenberg@unige.ch

**Competing interests:** The authors declare that no competing interests exist.

## Introduction

Cellular adaptation to a changing local environment is imperative for survival and proliferation. This is effected through a collection of sensing and signaling pathways that integrate information about the local environment and induce the requisite changes in various cellular programs that control organelle abundance and function, through multiple routes, including the modulation of transcription. In recent years, several transcriptional 'master regulators' of organellar biogenesis have been reported for mitochondria (*Jornayvaz and Shulman, 2010*), autophagosomes (*Kang et al., 2012*; *Chauhan et al., 2013*) and lysosomes (*Sardiello et al., 2009*). While the functional details of this control are still under investigation, coordinated transcriptional control of specific organelles is an emerging theme in cell biology.

Lipid droplets are the primary storage organelle for neutral lipids in the cell (*Meyers et al., 2017*). Intriguingly, the number and nature of these organelles vary greatly, both over time within a cell, and between cell types (*Thiam and Beller, 2017*). While a major function of lipid droplets is clearly as the storehouse of triglycerides and sterol esters, the diversity and variation of this organelle likely reflect the number of reported alternate functions of lipid droplets such as regulation of inflammation, general metabolism, and host-pathogen interplay (*Barisch and Soldati, 2017*; *Melo and Weller, 2016*; *Konige et al., 2014*). Despite the recognized importance of this organelle in health and disease, little is known of the signaling systems or proximal transcriptional regulators that control lipid droplet biogenesis, function and turnover in cells.

Recently, we used genome-wide, high-content siRNA screens to identify genes that affect cellular lipids. This analysis revealed that the Wnt ligand can potently stimulate lipid droplet accumulation in multiple cell types (*Scott et al., 2015*). In this paper, we report that the biogenesis of lipid droplets induced by Wnt signaling does not depend on the canonical TCF/LEF transcription factors. Our data show that the pro-lipid droplet signal induced by Wnt3a is mediated by members of the TFAP2 family of transcription factors. We thus conclude that TFAP2 may function as a 'master' regulator of lipid droplet biogenesis.

## Results

Wnt-induced lipid droplet formation could be conveniently visualized using BODIPY, which accumulated in lipid droplets (*Figure 1A*) and (*Scott et al., 2015*), and quantified by automated microscopy (*Figure 1B*, and all subsequent figures). Similarly, accumulation of lipid droplets in Wnt-treated cells could also be revealed in cells expressing the lipid droplet protein PLIN1a tagged with the GFP (*Figure 3—figure supplement 1C–F*; quantification in D and G, respectively) or by direct determination of triglyceride and cholesteryl ester amounts (*Figure 3—figure supplement 1*) and (*Scott et al., 2015*). In our previous work, we had observed that Wnt stimulates lipid droplet accumulation through upstream elements of the Wnt signalling pathway, including the canonical surface receptors and adenomatous polyposis coli (APC), a component of the destruction complex (*Scott et al., 2015*). This role of Wnt is well in-line with the established functions of Wnt signaling in the control of cellular metabolism, including carbohydrate, protein and lipid (*Prestwich and Macdougald, 2007*; *Sethi and Vidal-Puig, 2010*; *Ackers and Malgor, 2018*). Therefore, to further characterize the signaling cascade leading to the accumulation of lipid droplets after Wnt addition, we tested the key components from the canonical Wnt signalling pathway for a role in lipid droplet regulation, starting with the core enzyme of the destruction complex, GSK3B (*Figure 1A–C*). We found that overexpression of both the wild-type, and the constitutively active S9A (*Stambolic and Woodgett, 1994*) mutant of GSK3B were capable of attenuating lipid droplet accumulation in response to Wnt3a-treatment (*Figure 1A–B*), consistent with the function of GSK3B activity as a negative regulator of Wnt signalling. Further, siRNAs to GSK3B were sufficient to induce lipid droplet accumulation (*Figure 1A,C*) — much like we had shown after gene silencing of APC, another member of the destruction complex (*Scott et al., 2015*).

We next investigated the possible role of the downstream targets of the Wnt pathway at the transcription level. Surprisingly, siRNAs to TCF/LEF transcription factors relevant in canonical Wnt signaling, failed to affect lipid droplet accumulation - and yet there is no doubt that lipid droplet accumulation in response to Wnt3a is transcriptionally mediated. Indeed, the expression of SOAT1 and DGAT2, which encode key-enzymes of lipid droplet formation, increased in response to Wnt3a-treatment (*Scott et al., 2015*). In addition, silencing these genes inhibited lipid droplet accumulation in response to Wnt3a, most potently in combination with each other (*Figure 1D*; silencing efficiency *Figure 1—figure supplement 1B*), or with specific inhibitors to DGAT1 or SOAT (*Figure 1E*, *Figure 1—figure supplement 1A*). Finally, Wnt3a-induced lipid droplets were decreased after treatment with the general inhibitor of transcription Actinomycin D (*Figure 1E*). While these data altogether confirmed that the Wnt pathway was mediating the pro-lipid droplet signal, our inability to link lipid droplet induction to TCF/LEF led us to consider the possibility that a branching signaling path, under the control of GSK3B and/or ß-catenin but not the canonical Wnt transcription factors TCF/LEF, was inducing the accumulation of lipid droplets in cells. To explore this possibility, we initiated several parallel and complementary systems-level analyses with the aim to identify the transcriptional regulators proximal to lipid droplet biogenesis.

To better understand the nature of the pro-lipid droplet signal induced by Wnt, we revaluated the involvement of individual components of the Wnt signaling pathway in the induction of lipid droplet accumulation. First, we tested the ability of each of the 19 human Wnt ligands to induce lipid droplet accumulation in L Cells by transfection and autocrine or paracrine induction of the Wnt pathway (*Figure 1F–G*). Wnt ligands displayed a broad, but not universal capacity to induce lipid droplet accumulation that paralleled both their evolutionary pedigree (*Figure 1F*), and previously reported abilities to activate canonical Wnt signaling as measured by a TCF/LEF reporter system (*Najdi et al., 2012*). This confirmed that the pro-lipid droplet signal was indeed transiting, at least initially, through canonical Wnt signaling components.

To systematically assess the involvement of the remaining components of the Wnt pathway for involvement in the lipid droplet response, we performed a targeted screen for factors influencing lipid droplet accumulation using a library of 73 compounds selected for known interactions with elements of the Wnt pathway (see Materials and methods). We tested the library in both, Wnt3a-stimulated conditions to assess any inhibitory activity of lipid droplet accumulation, and unstimulated conditions to identify compounds with the ability to induce the phenomenon (*Figure 1—source data 1*). Indeed, treatment with several compounds reported to activate the Wnt-pathway-induced lipid droplet accumulation, such as BML-284 (activator of ß-catenin [*Liu et al., 2005*]), doxorubicin

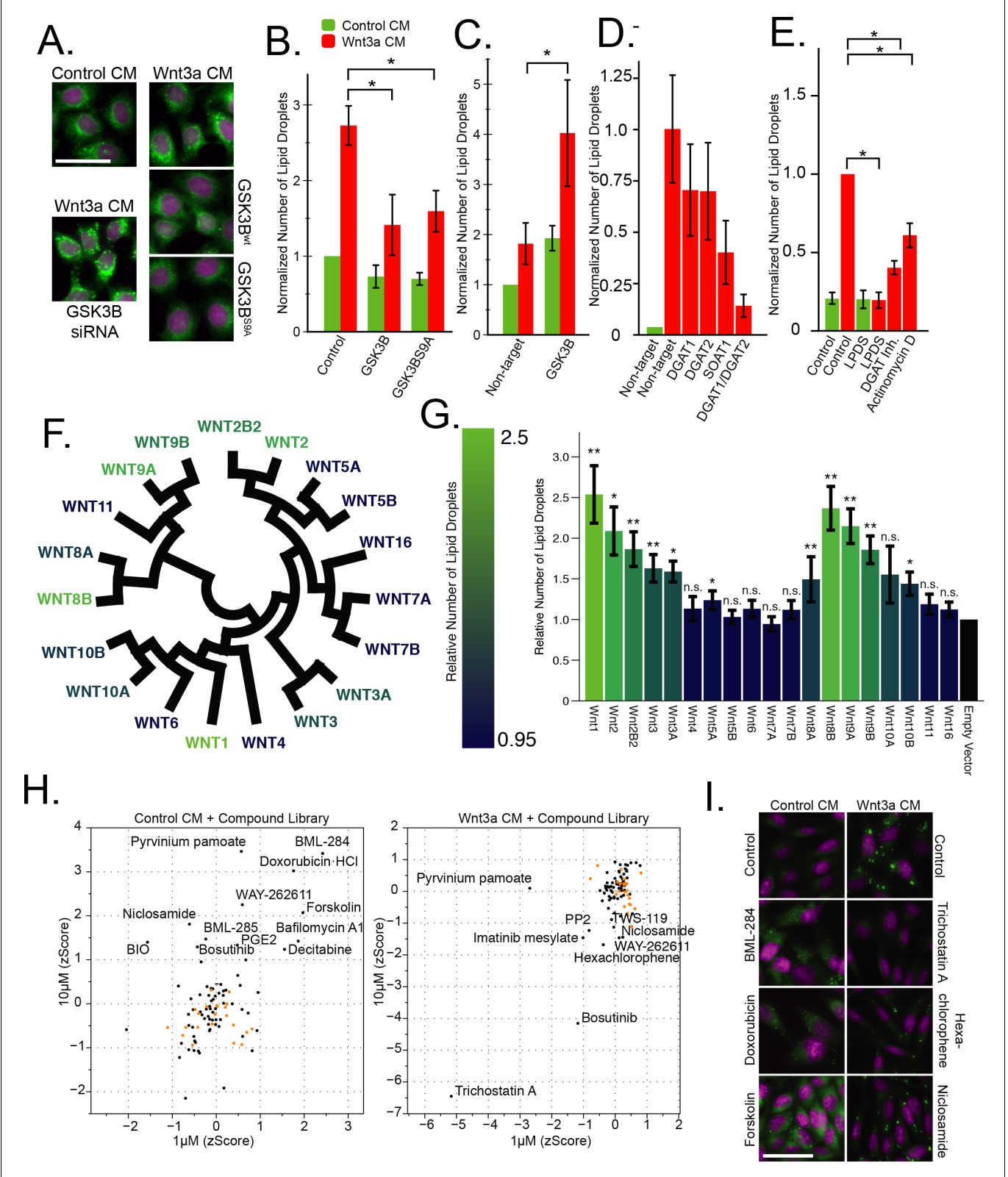

**Figure 1.** The Wnt pathway and the regulation of lipid droplets. (A–B) HeLa-MZ cells were transfected with plasmids encoding wild-type or a S9A mutant of GSK3B 24 hr before the addition of Wnt-3a- or control-conditioned media (CM) for a further 24 hr. Cells were fixed, labeled with BODIPY (lipid droplets, green) and Hoechst 33342 (nuclei, magenta), and imaged by light microscopy. In (B) the number of lipid droplets was quantified by automated microscopy (bar graph), and the data are presented as the mean number of lipid droplets per cell of five independent experiments ± SEM,

*Figure 1 continued on next page*

*Figure 1 continued*

normalized to the control condition. (**C–D**) As in (**A**), except that HeLa-MZ cells (**C**) or L Cells (**D**) were transfected with siRNAs against the indicated targets for 48 hr, before the addition of Wnt3a. Efficient silencing was confirmed by qPCR (*Figure 1—figure supplement 1B*) and the data are presented as the mean number of lipid droplets per cell of 3 independent experiments ± SEM, normalized to the control condition. (**E**) L cells were incubated with the indicated compounds together with Wnt3a for 24 hr, processed and analyzed as in (**A**) and the data are presented as the mean number of lipid droplets per cell of five independent experiments ± SEM, normalized to the control condition. (**F**) Evolutionary relationship of the 19 Wnt ligands. Color indicates ability to induce lipid droplets as detailed in (**G**). (**G**) L Cells were transfected with plasmids containing each of Wnt ligand for 48 hr, imaged and analyzed as in (**A**). Data are normalized to the empty vector control and were tested for significance and are presented as the mean number of lipid droplets per cell of two independent replicates of the screen ± SEM, normalized to the control condition. The data are color-coded from a high (light) to a low (dark) number of lipid droplets induced by each Wnt ligand. (**H–I**) High-content image-based screen of a library of compounds that affect the Wnt pathway in HeLa-MZ cells. Cells were incubated for 24 hr with Wnt-3a- or control-conditioned media for 24 hr in the presence of the compounds at 1 µM and 10 µM, fixed, labeled with BODIPY (lipid droplets) and Hoechst 33342 (nuclei) and imaged by automated microscopy. The number of droplets per cell was counted and the zscores established, in order to quantify the ability of each compounds to induce lipid droplets in untreated cells (H, left panel), or to inhibit lipid droplet formation in Wnt3a-treated cells (H, right panel). Panel I illustrates the effects of compounds that induce droplet formation (left column) or that do (Trichostatin A) or do not (niclosamide, hexachlorophene) inhibit droplet formation (right column) in Wnt3a-treated cells. Nuclei are in magenta, and lipid droplets in green. Green bars, control-conditioned media (control CM); Red bars, Wnt3a-conditioned media (Wnt3a CM). In this figure, pValues are indicate as: *,<0.05; **, <0.005, and n.s., not significant.

DOI: https://doi.org/10.7554/eLife.36330.002

The following source data and figure supplement are available for figure 1:

**Source data 1.** Effect of Wnt pathway related compounds on lipid droplet induction.

DOI: https://doi.org/10.7554/eLife.36330.004

**Figure supplement 1.** Lipid droplet accumulation in response to Wnt3a: combinatorial treatments against lipid droplet enzymes by RNAi and chemical inhibitors.

DOI: https://doi.org/10.7554/eLife.36330.003

(activator of Wnt signalling [*Dai et al., 2009*]), and forskolin (activation via PKA-Wnt crosstalk [*Zhang et al., 2014*]) (*Figure 1H–I*). Conversely, known inhibitors of the pathway such as Tricostatin A (epigenetic regulator of DKK1 [*Vibhakar et al., 2007*]) hexachlorophene (ß-catenin inhibitor [*Park et al., 2006*]), and niclosamide (inducer of LRP6 degradation [*Lu et al., 2011*]) significantly decreased the appearance of lipid droplets in response to Wnt3a (*Figure 1H–I*).

While these data certainly confirm the role of Wnts in regulating lipid droplets, they did not reveal the pathway linking the Wnt destruction complex to the transcriptional changes we observed (*Figure 1A–E*, [*Scott et al., 2015*]). Given these results, and the large number of ß-catenin-independent targets of the destruction complex (*Kim et al., 2009*), we initiated several strategies to identify candidate regulators, in particular the proximal transcription factors directly upstream from lipid droplet biogenesis. Our aim was to identify factors linked to Wnt signalling and to characterize the signalling pathway from ligand-stimulation to lipid droplet accumulation.

We first started by taking the subset of genes annotated as 'transcription factor activity' (GO:0000988) from our primary genome-wide siRNA screen data (*Scott et al., 2015*) to identify transcription factors that influenced cellular cholesterol levels in the cell (*Figure 2—figure supplement 1B*; *Figure 2—source data 1*). While several of these candidates have well-established roles in regulating general proliferation (i.e. MACC1, JDP2, SP3, TP53, ZNF217, TAF1), or links to the WNT pathway in keeping with our findings (i.e. GLI3, SIX2, SOX9, FOXK2, BARX1), we were particularly interested in identifying candidates with reported functions in lipid metabolism. The latter subset included ARNT2, STAT3, KDM3A, ATF5, KLF5, KLF6 and members of the TFAP2 (AP-2) transcription factor family (see below).

As a second approach to search for candidate transcriptional regulators of lipid droplets, we compared existing transcriptome data of Wnt3a-treated cells with that of other conditions known to induce the accumulation of lipid droplets in tissue culture cells. It is well-established that the formation of lipid droplets is stimulated artificially by the addition of exogenous fatty acids (*Martin and Parton, 2006*), and the process has been studied at the transcriptional level in multiple studies. We therefore combined our Wnt3a gene array data (*Scott et al., 2015*) with three published datasets of mRNA levels after treatment of cells with fatty acids or knockout of lipid droplet regulatory factors (*Li et al., 2010*; *Lockridge et al., 2008*; *Shaw et al., 2013*). Our rationale was to identify the relevant transcription factors required for the induction of lipid droplet biogenesis by inferring from the expression data what is the common set of transcription factors active in response to Wnt3a and/or

to the modulation of lipid droplets by fatty acid treatment or gene knockout. This analysis involved testing for over-representation of genes annotated to be regulated by a specific transcription factor in the set of the most perturbed genes after either treatment. Wnt3a-stimulation influenced a larger number of transcriptional regulators (172) as compared to lipid droplet modulation (91), but the vast majority of this subset (>75%) were also changed by Wnt3a (*Figure 2—figure supplement 1B*). With this approach, we found that many candidate transcription factors known to function in both lipid metabolism and adipogenesis were influenced by Wnt3a-treament and fatty acid perturbation (*Figure 2—source data 2*), including TFAP2A (p-value Wnt3a treatment: $3.5 \times 10^{-9}$; p-value fatty acid perturbation: $4.8 \times 10^{-4}$).

As a third systems-level approach, we undertook a direct examination of the promoter regions of annotated lipid droplet proteins as was used by Sardiello and colleagues to identify the transcription factor TFEB and the CLEAR element as master regulators of lysosome biogenesis (*Sardiello et al., 2009*). We collected and examined the upstream promoter sequences for the 145 proteins annotated as 'Lipid Droplet' (GO:0005811; *Figure 2—source data 3*) and tested for over-represented sequence motifs. Among the most enriched motifs in the upstream promoter region of lipid droplet genes were motifs identified as TFAP2A (pValue: $9.0 \times 10^{-3}$) and TFAP2C (pValue: $1.8 \times 10^{-5}$) binding sites (*Figure 2—figure supplement 2A*). Further analysis revealed that 74 of the 'Lipid Droplet' proteins contained at least one of the TFAP2A or TFAP2C annotated binding sites (*Figure 2—figure supplement 2B*, *Figure 2—source data 3*), which were generally present within the first few hundred base-pairs from the start site (*Figure 2—figure supplement 1C*).

The TFAP2 (AP-2) family of basic helix-span-helix transcription factors have been long recognized to play key roles during development (*Eckert et al., 2005*). Yet, little else is known regarding their function in adult animals where these proteins are expressed, although various TFAP2 homologs have been linked to tumour progression in cancer models (*Eckert et al., 2005*; *Li and Dashwood, 2004*; *Li et al., 2009*). The family consists of five proteins in human and mouse, and are thought to form homo- and heterodimers that bind to similar promoter sequences albeit with different affinities (*Eckert et al., 2005*). Given our observation that downstream genes known to be regulated by TFAP2 family members change in response to Wnt3a (*Figure 2—figure supplement 1B*), that the silencing of TFAP2 genes can regulate cholesterol amounts in cells (*Figure 2—figure supplement 1A*), and that TFAP2 consensus sites are over-enriched in the promoter sequences of genes encoding for lipid droplet proteins (*Figure 2—figure supplement 2A*), we began to suspect that TFAP2 proteins mediate the pro-lipid droplet signal induced by Wnt3a. This notion was further buttressed by previous studies showing that TFAP2A directly interacts with both ß-catenin and APC (*Li et al., 2009*; *Li et al., 2015*) making this family of transcription factors our leading candidate for mediating the pro-lipid droplet signalling activity of Wnt3a.

To gain a detailed description of the transcriptional changes in the context of TFAP2, Wnt and lipid droplets, we performed an RNAseq determination of mRNA levels in cells treated with Wnt3a for short times (2 hr and 6 hr) with the aim to identify early factors of the transcriptional control relevant for lipid droplet biogenesis. As expected, a pathway analysis of the most responsive genes at 2 hr found over-representation of terms related to mRNA processing, DNA binding and transcriptional regulation (*Figure 2A*), consistent with the expected nature of the early Wnt3a-responsive genes. By 6 hr post-Wnt3a treatment, transcriptional regulators were still over-represented, but additional terms reflecting downstream effector pathways were present, including those related to glucose metabolism and endosomal trafficking, as well as fatty acid and cholesterol related genes (*Figure 2A*) consistent with our previous findings (*Scott et al., 2015*). Indeed, the RNAseq analysis found that SREBF1 (Sterol Regulatory Element Binding Transcription Factor 1) mRNA levels were the most decreased of any transcription factor (0.60 of control) at the later time point (*Figure 2B*).

Intriguingly, the most upregulated transcription factor at early times was DDIT3 (DNA Damage Inducible Transcript 3, also known as CHOP) — and increased DDIT3 could readily be detected at the mRNA and protein level (*Figure 2—figure supplement 3A—B*). This member of the CCAAT/ enhancer-binding (CEBP) protein family is a potent and direct inhibitor of SREBF1 transcription (*Chikka et al., 2013*) and a known regulator of cellular lipid metabolism. While this interaction may contribute to the decrease in cellular free cholesterol and the downregulation of cholesterol metabolic enzymes after Wnt3a treatment (*Figure 2A*, [*Scott et al., 2015*]), overexpression of constitutively active SREBF1 truncations (*Shimano et al., 1997*) had essentially no effect on the formation of lipid droplets, whether Wnt3a was present or not (*Figure 2—figure supplement 4A*)

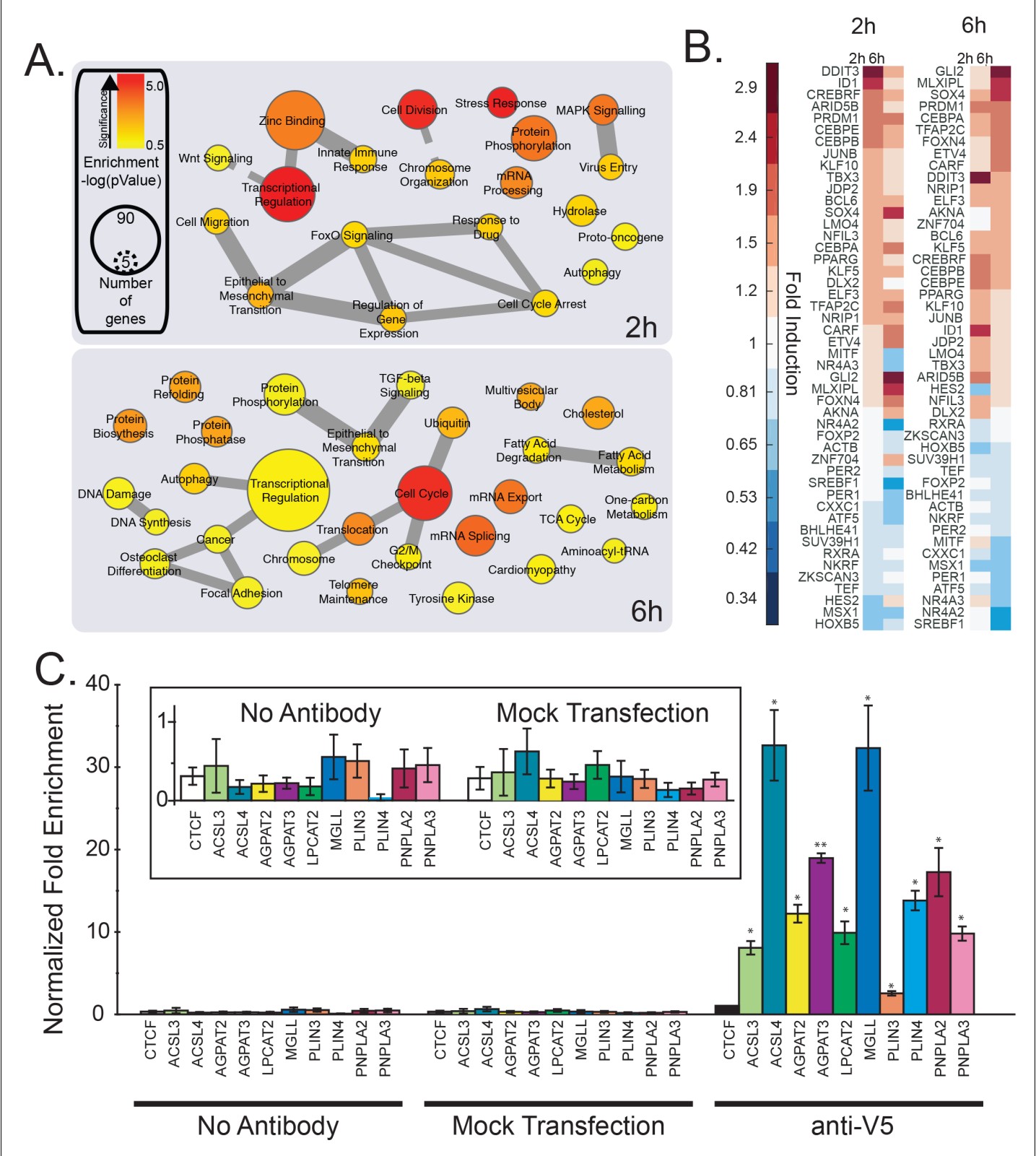

**Figure 2.** mRNA profiling and analysis of gene expression of cells treated with Wnt3a. (**A-B**) HeLa-MZ cells were treated with control- or Wnt3a-conditioned media for 2 hr or 6 hr before RNA isolation and RNAseq analysis. Panel (A) shows the pathway enrichment of perturbed mRNAs. Node size indicates number of genes in each ontology and colour the statistical strength of the enrichment. Edge thickness indicates the strength of overlap of related ontologies. From (A), the fold change of transcription factors amounts in response to Wnt3a is shown in panel B. C. The ability of TFAP2 family

*Figure 2 continued on next page*

*Figure 2 continued*

member TFAP2A to bind to regulatory regions of lipid droplet at lipid metabolic enzyme genes was tested by ChIP-qPCR (see Materials and methods). Data are presented as the mean DNA amounts normalized to the negative control (CTCF) of three independent experiments ± SEM. (*) indicates a p-value<0.05; (**) indicates a p-value<0.005. Inset; re-scaled view of signal of the control conditions.

DOI: https://doi.org/10.7554/eLife.36330.005

The following source data and figure supplements are available for figure 2:

**Source data 1.** Effect of silencing transcription factors on cellular cholesterol amounts.

DOI: https://doi.org/10.7554/eLife.36330.010

**Source data 2.** Comparative enrichment of transcriptional targets in cells treated with Wnt3a or fatty acid perturbation.

DOI: https://doi.org/10.7554/eLife.36330.011

**Source data 3.** TFAP2 family member consensus binding sites in lipid droplet genes.

DOI: https://doi.org/10.7554/eLife.36330.012

**Figure supplement 1.** Datamining for putative lipid droplet transcriptional regulators.

DOI: https://doi.org/10.7554/eLife.36330.006

**Figure supplement 2.** The consensus binding sites of TFAP2 family members are overrepresented in lipid droplet genes.

DOI: https://doi.org/10.7554/eLife.36330.007

**Figure supplement 3.** Effect of Wnt3a on DDIT3 protein and mRNA amounts in L Cells.

DOI: https://doi.org/10.7554/eLife.36330.008

**Figure supplement 4.** SREBF activity does not significantly influence lipid droplet number.

DOI: https://doi.org/10.7554/eLife.36330.009

despite activating known target genes involved in cholesterol metabolism (*Figure 2—figure supplement 4C*). Neither did treatment with the S1P/SREBF1 inhibitor PF-429242 (*Hawkins et al., 2008*) (*Figure 2—figure supplement 4B*) in either Wnt3a-stimulated, or naive conditions. These observations suggest that while relevant to the observed changes in cellular cholesterol homeostasis generated by Wnt3a, putative DDIT3-induced changes in SREBF1 expression have no significant role in lipid droplet accumulation in response to Wnt3a.

Along with DDIT3 and SREBF1, our list of early Wnt3a-responsive transcription factors includes several that have known functions in regulating cellular lipid homeostasis such as CEBPB and CEBPE, KLF5, KLF10, PPARG, MLXIPL, and PER2. Our list also included the TFAP2 family member TFAP2C, which our datamining strategies had already identified as involved in lipid homeostasis, and as a candidate transcription factor controlling lipid droplet biogenesis. In fact, 6 hr post-Wnt3a stimulation TFAP2C (1.72-fold) was among the most upregulated transcription factors (*Figure 2B*).

Given that our datamining efforts identified TFAP2 family members as putative transcription factors regulating lipid droplet proteins and that TFAP2C was among the most upregulated transcription factors in response to Wnt3a (*Figure 2B*), we next investigated whether TFAP2 family members played a direct role in regulating lipid droplets. To this end, we tested the ability of TFAP2A to directly bind to the promoter region of known lipid droplet, and lipid metabolic genes containing a predicted TFAP2 consensus site by ChIP-PCR (*Figure 2C*), including the enzymes ACSL3, ACSL4, AGPAT2, AGPAT3, LPCAT2, and MGLL, and the lipid droplet resident proteins PLIN3, PLIN4, PNPLA2, and PNPLA3 (*Meyers et al., 2017*; *Barneda and Christian, 2017*). Indeed, we found that the TFAP2A protein was able to bind the upstream promoter of all the genes we tested, supporting the notion that expression was controlled by TFAP2 family members. Next, we tested whether Wnt3a retained the ability to induce lipid droplets after TFAP2 depletion by RNAi. While knockdown of either TFAP2A or TFAP2C had no or only a modest effect, tandem silencing of both homologs produced a marked reduction in the number of lipid droplets present in cells in response to Wnt3a (*Figure 3A–B*; knock-down efficiency *Figure 3—figure supplement 2*) and markedly decreased the amounts of cholesteryl esters (*Figure 3—figure supplement 1A*). In keeping with these findings, these siRNA treatments diminished the mRNA levels of SOAT1, a key enzyme proximal to the production of lipid droplets (*Figure 3C*) that mediate the production of cholesteryl esters (*Chang et al., 2001*). These observations indicate that TFAP2 family members exhibit complementary functions.

As an alternative approach, we used CRISPR/Cas9 gene knockout to generate HeLa-MZ cells clones lacking TFAP2A (*Figure 3D–E*). While tandem depletion by RNAi was necessary to reduce lipid droplet production after Wnt3a addition in L Cells, two knockout clones of TFAP2A

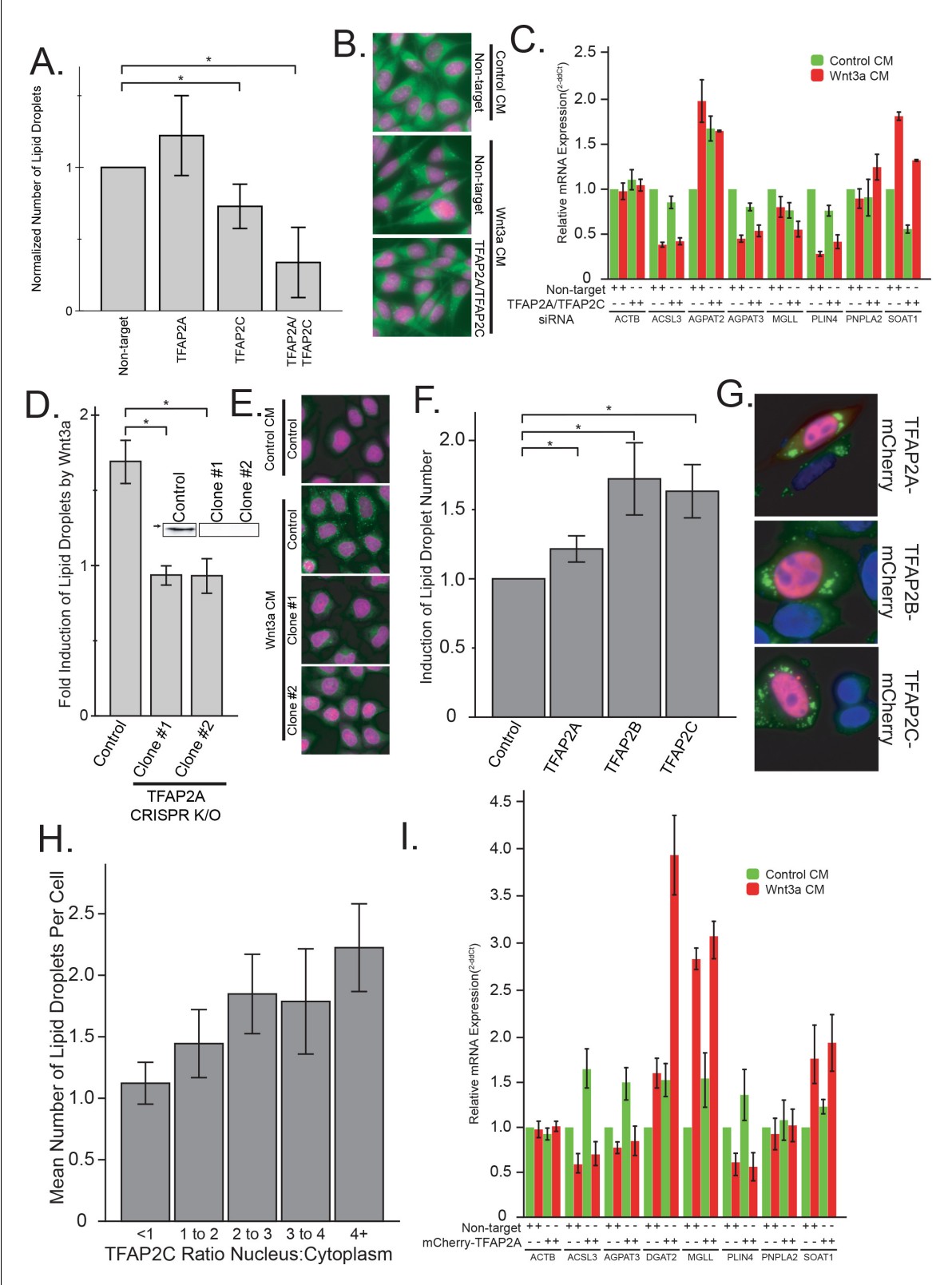

**Figure 3.** The TFAP2 family of transcription factors are both necessary and sufficient to mediated lipid droplet accumulation. (A–B) L Cells were treated with siRNAs against the indicated targets for 48 hr before the addition of Wnt3a-conditioned medium for an additional 24 hr. Cells were then fixed, labeled, imaged and analyzed by automated microscopy as in *Figure 1A*. In (A), data are presented as the normalized mean number of lipid droplets per cell of 5 independent experiments ± SEM. Cells treated with non-target siRNAs or with siRNAs to both TFAP2A and TFAB2C are shown in panel B

*Figure 3 continued on next page*

*Figure 3 continued*

(nuclei in magenta; lipid droplets in green). (**C**) L cells were treated with siRNAs against both TFAP2A and TFAP2C or with non-targeting controls as in (**A**), before the addition of Wnt3a- or control-conditioned media for an additional 24 hr. RNA was isolated and analyzed by qPCR using primers to the indicated genes, and that data are expressed relative to the non-target control and are presented as the mean mRNA amounts of two to five independent experiments ± SEM. (**D–E**) HeLa-MZ cells were transfected with targeted CRISPR/Cas9 plasmids against TFAP2A. The corresponding knock-out clones as well as control cells were treated with Wnt3a-conditioned media for 24 hr. In **D**), the number of lipid droplets was quantified as in *Figure 1A* and is expressed as fold induction relative to the control cells in five independent experiments ± SEM. Inset: TFAP2A protein levels of each clone determined by Western blot. Arrow indicate position of 50 kDa marker. Representative images are shown in E (nuclei in magenta; lipid droplets in green). (**F–H**) L Cells were transfected or not with mCherry-tagged TFAP2 family members for 48 hr before fixation, labeling and imaging as in *Figure 1A*. The mean number of lipid droplets per cell expressing each mCherry-tagged TFAP2 protein was counted, and is expressed, as in panel (**D**), as fold induction relative to the control cells in six independent experiments ± SEM. Panel G shows cells expressing each mCherry-tagged TFAP2 protein (Blue, nucleus; Green, lipid droplets; Red, TFAP2-mCherry fusion proteins), and panel H shows the number of lipid droplets per cell in cells overexpression TFAP2C-mCherry, binned by their nuclear:cytoplasmic distribution. Data are the mean lipid droplets per cell ±SEM for 450 cells. (**I**) L Cells were treated as in (**F**) before extraction and determination of the indicated mRNAs by qPCR. Data are presented as the mean mRNA amounts of two to five independent experiments ± SEM. In this figure, (*) indicates a p-value<0.05.
DOI: https://doi.org/10.7554/eLife.36330.013

The following figure supplements are available for figure 3:

**Figure supplement 1.** Characterization of lipid droplets by PLIN1a-GFP, cholesterol esters and triglycerides.
DOI: https://doi.org/10.7554/eLife.36330.014
**Figure supplement 2.** Characterization of knock-down efficiencies and relative over-expression levels by qPCR.
DOI: https://doi.org/10.7554/eLife.36330.015

demonstrated a complete lack of change in lipid droplet number after Wnt3a stimulation (*Figure 3D–E*; and see *Figure 3—figure supplement 1C–D*) and failed to accumulate triglycerides (*Figure 3—figure supplement 1B*). The ability of Wnt to induce LD formation could be rescued upon TFAP2A re-expression (*Figure 4—figure supplement 1D*), demonstrating that the inhibition observed in knock-out cells was not caused by some off-target or indirect effect of the treatment. Together, these results imply that TFAP2A/TFAP2C are necessary for mediating the pro-lipid droplet signal of the Wnt pathway.

We next sought to determine if the TFAP2 family is sufficient for the induction of lipid droplets. For this, we fused full-length TFAP2A, TFAP2B, and TFAP2C to mCherry and overexpressed these as exogenous chimeras. As transcription factors the TFAP2 family members function in the nucleus, the mCherry-tagged TFAP family members exhibited a somewhat heterogeneous distribution between cytoplasm and nuclei, presumably because of variations in expression levels. Impressively, the number of lipid droplets per cell increased with increased nuclear localization of each TFAP2 family member (*Figure 3H*; (*Figure 3—figure supplement 1F–G*); relative overexpression *Figure 3—figure supplement 2B*), and, in cells with TFAP2 nuclear localization, the expression of each family member was clearly sufficient to cause lipid droplet biogenesis (*Figure 3F–G*). Moreover, the expression of TFAP2C was also able to trigger the expression of lipid droplet enzymes (*Figure 3I*) and accumulation of cholesterol esters and triglycerides (*Figure 3—figure supplement 1E*), further supporting the notion that TFAP2 family members function as transcriptional regulators of lipid droplet biogenesis.

In totality, these findings demonstrate TFAP2A, TFAP2B and TFAP2C are sufficient to induce biogenesis of lipid droplets when expressed in cells, and members of the TFAP2 family are required to mediate the accumulation of lipid droplets seen in response to Wnt-stimulation. This finding is in keeping with a previous report that targeted overexpression of TFAP2C in mouse liver induced steatosis, accumulation of fat, and eventual liver failure (*Holl et al., 2011*), and given the enrichment in TFAP2 binding sites in lipid droplets proteins (*Figure 2—figure supplement 2*), implicate the TFAP2 family as central regulators of lipid droplet biogenesis.

Given our observation that TFAP2C expression correlates with DDIT3 expression in cells after Wnt3a treatment (*Figure 2B*), we investigated the role of DDIT3 in control of lipid droplets. We started by examining the promoter region of the transcription factors whose mRNA levels were impacted by Wnt3a-treatment (*Figure 2B*) for reported DDIT3::CEBPA consensus sites (*Ubeda et al., 1996*). Intriguingly, not only did we find such a site upstream of the TFAP2C gene, but there was a DDIT3::CEBPA consensus site in 7 out of the 10 lipid-related transcription factors identified in our RNAseq (*Figure 4—figure supplement 1A*), a 1.8-fold enrichment over the

frequency in the total Wnt3a-influenced transcription factor set. This lends further support for the view of DDIT3 as an important transcriptional regulator of lipid homeostasis in cells (*Chikka et al., 2013*).

To address this notion directly, we next tested whether DDIT3 itself was able to influence both TFAP2 family members, and the number of lipid droplets in cells. Indeed, overexpression of DDIT3-mCherry chimera increased mRNA levels of lipid droplet enzymes, without much of an effect on lipid droplet coat proteins (*Figure 4A*; relative overexpression *Figure 3—figure supplement 2D*), with a concomitant increase in cholesteryl esters and triglycerides (*Figure 3—figure supplement 1B*). Further, silencing of DDIT3 influenced the mRNA levels of many lipid metabolic genes including decreasing the levels of SOAT1 (*Figure 4B*; knock-down efficiency *Figure 3—figure supplement 2C*), as well as reducing cholesteryl ester accumulation (*Figure 3—figure supplement 1A*) and lipid droplet accumulation (*Figure 4C*) in response to Wnt3a. These observations suggest that DDIT3 plays a regulatory role in lipid droplet biogenesis, presumably in concert with TFAP2.

We next tested the requirement of DDIT3 for the lipid droplet response directly by interfering with DDIT3 expression by both RNAi and CRISPR/Cas9. Silencing of DDIT3 expression diminished accumulation of lipid droplets both in response to Wnt3a stimulation (*Figure 4D*; *Figure 4—figure supplement 1B*; *Figure 3—figure supplement 1C–D*), and after silencing of APC (*Figure 4—figure supplement 1B*), an alternative strategy previously shown to be sufficient to induce lipid droplets (*Scott et al., 2015*). Wnt3a-induced increase in triglycerides was also ablated in these CRISPR clones (*Figure 3—figure supplement 1B*). Further, knock-out clones lacking DDIT3 were completely non-responsive to Wnt3a with regards to lipid droplet number (*Figure 4D*). Much like with TFAP2A, DDIT3 re-expression in knock-out cells rescued Wnt-induced LD formation (*Figure 4—figure supplement 1D*), demonstrating that the inhibition observed in KO cells was not due to some off-target effects. In the context of the significant increase in DDIT3 message and protein (*Figure 2—figure supplement 3*), these results suggest that induction of DDIT3 transcription in response to Wnt3a is necessary for lipid droplet biogenesis.

Given that the presence of DDIT3 was necessary to convey the pro-lipid droplet signal, we next tested whether over-expression of the transcription factor is sufficient to induce lipid droplet accumulation. As with TFAP2, overexpression of mCherry-tagged fusions of DDIT3 was sufficient to increase lipid droplet numbers (*Figure 4E*; *Figure 4—figure supplement 1C*; *Figure 3—figure supplement 1F–G*) as well as cholesterolyl esters and triglyceride levels (*Figure 3—figure supplement 1E*) in transfected cells as compared to the control. In total, these data suggest that in addition to regulation of cholesterol metabolism (*Chikka et al., 2013*), DDIT3 may function as a more global regulator of cellular lipid homeostasis in part through regulation of the TFAP2 family of transcription factors. In an attempt to clarify the relationship between DDIT2 and TFAP2A, we tested whether TFAP2A overexpression restored Wnt-dependent LD formation in DDIT3 knock-out cells and vice versa (*Figure 4—figure supplement 1D*). While TFAP2A did not rescue DDIT3 knock-out cells, DDIT3 expression was able to restore LD formation in TFAP2A knock-out cells. Although future work will be required to determine the precise functions of DDIT3, it is tempting to speculate that TFAP2A drives DDIT3 expression (or that DDIT3 responds lipid droplet accumulation). DDIT3 therefore, in addition to a direct inducer of lipid storage, might act as TFAP2A repressor — in line with the known repressor function of DDIT3 in SREBP expression (*Chikka et al., 2013*).

## Discussion

In conclusion, our data show TFAP2 family members function to modulate expression of lipid droplet proteins and induce the accumulation of lipid droplets in cells. We found TFAP2 is necessary to potentiate the pro-lipid droplet signal induced by Wnt3a, and expression of TFAP2 family members is sufficient to induce lipid droplet accumulation in cells (*Figure 3*).

Not only do these data support the view that the TFAP2 family of transcription factors can function as regulators of lipid droplet biogenesis, they provide insight into the transcriptional network directing changes in lipid homeostasis and the accumulation of lipid droplets in response to Wnt stimulation. Our data also provides further decoding of the known relationships between Wnt signaling and the control of cellular metabolism (*Prestwich and Macdougald, 2007*; *Sethi and Vidal-Puig, 2010*; *Ackers and Malgor, 2018*). Finally, our observations indicate that, in addition to the canonical TCF/LEF transcriptional response, Wnt signalling via APC, GSK3, and ß-Catenin induces

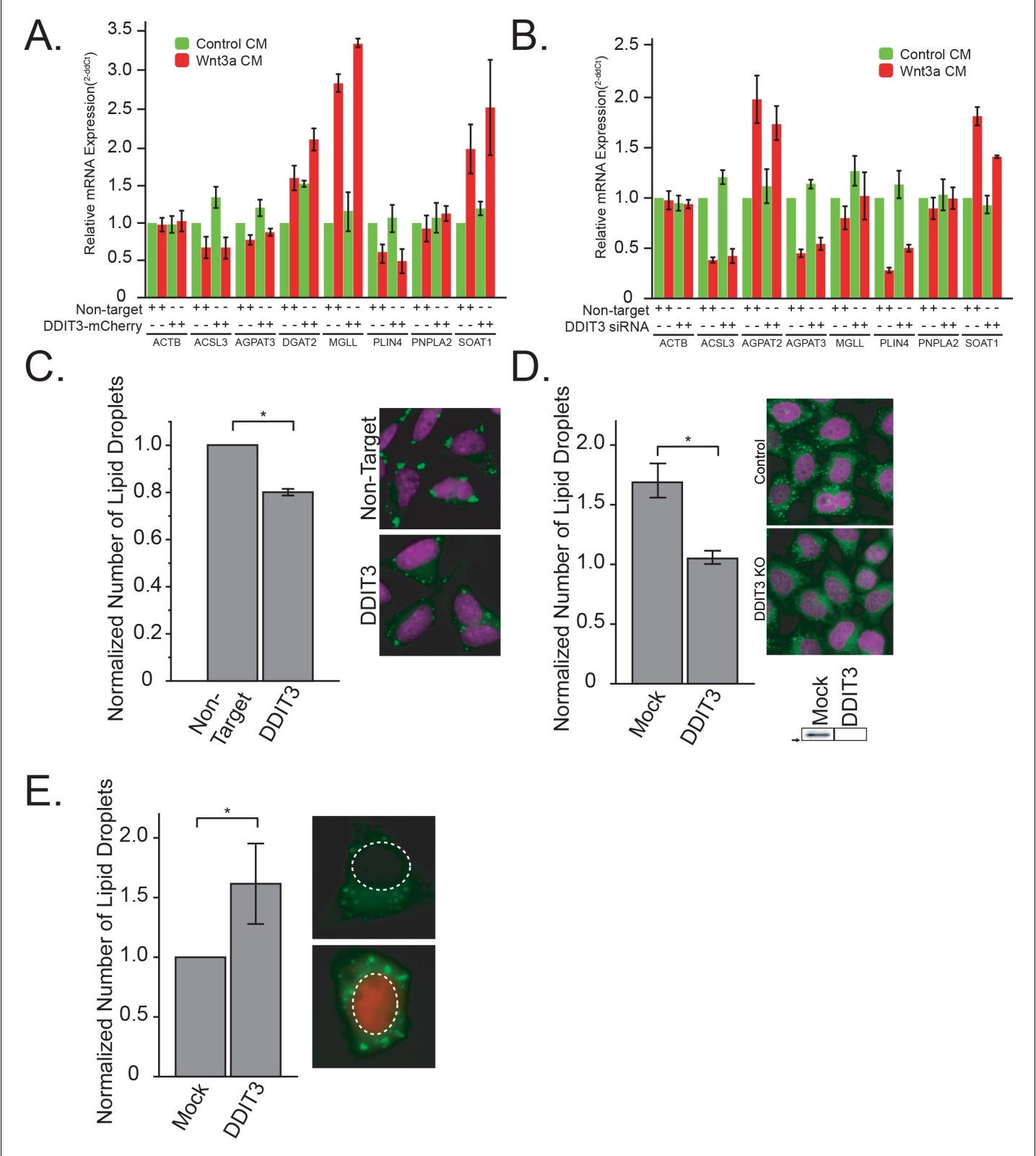

**Figure 4.** DDIT3 is both necessary and sufficient to mediated lipid droplet accumulation A-B. qPCR of DDIT3, DGAT2 and SOAT1 after overexpression of DDIT3-mCherry (**A**), or siRNAs to DDIT3 (**B**). Data are presented as the mean of two to five independent experiments ± SEM. (**C**). L cells were treated, processed and analyzed like in *Figure 3A*, except that they were transfected with siRNAs to DDIT3 before stimulation with Wnt3a. Data are presented as the normalized mean number of lipid droplets per cell of three independent experiments ± SEM. (*) indicates a p-value<0.05. Panel E

*Figure 4 continued on next page*

*Figure 4 continued*

shows cells treated with non-target or anti-DDIT3 siRNAs (magenta, nucleus; green, lipid droplets). D. HeLa-MZ CRISPR/Cas9 clones were prepared and analysed as in *Figure 3D*. Data are expressed as fold induction relative to the control cells in five independent experiments ± SEM. Inset: DDIT3 protein levels of each clone determined by western blot. Arrow indicate position of 25 kDa marker. Representative images are shown (nuclei in magenta; lipid droplets in green). (E) L cells were treated, processed and analyzed like in *Figure 3F*, except that they were transfected with a plasmid encoding DDIT3-mCherry. Data are presented as the normalized mean number of lipid droplets per cell of six independent experiments ± SEM. (*) indicates a p-value<0.05. Panel E shows cells expressing or not DDIT3-mCherry (blue, nucleus; green, lipid droplets, red, DDIT3-mCherry fusion protein). In this figure, (*) indicates a p-value<0.05.

DOI: https://doi.org/10.7554/eLife.36330.016

The following figure supplement is available for figure 4:

**Figure supplement 1.** DDIT3 and lipid homeostasis.

DOI: https://doi.org/10.7554/eLife.36330.017

transcription-mediated lipid changes. These include decreased levels of SREBF1, which is likely to contribute to the observed reduction in total membrane cholesterol levels (*Scott et al., 2015*), and increased expression of DDIT3 and TFAP2, which triggers lipid droplet biogenesis with storage of cholesterol esters and triglycerides. These observations also lead to the notion that via TFAP2 transcription factors, Wnt exerts pro-proliferative effects in developmental and in pathological contexts by leading to the accumulation of lipid droplets.

In addition, Wnt3a has been recently identified as an intra-cell synchronizer of circadian rhythms in the gut, controlling cell-cycle progression, and the mRNA of Wnt3a itself exhibits circadian oscillations under the control of the master clock regulators Bmal1 and Per (*Matsu-Ura et al., 2016*). Given this function, it is tempting to speculate that Wnts serve a more general role as a mid- or long-range circadian signalling intermediates, linking lipid metabolism to the master transcriptional clocks that coordinate metabolic functions with the day-night cycle. Because of its circadian nature, and capacity to induce lipid droplet accumulation in a broad range of cells types, Wnts could serve as a fundamental signal for cells to store lipids during times when nutrients are expected to be in excess for later use during periods of rest or fasting.

Both lipid droplets enzymes (*Solt et al., 2012*), and the volume of lipid droplets themselves (*Uchiyama and Asari, 1984*), vary in a circadian fashion, which is completely consistent with a role as neutral lipid storage sites. TFAP2-binding sites are over-represented in the promoters of circadian-controlled genes (*Bozek et al., 2009*), suggesting that the Wnt/TFAP2 control of lipid droplets is one mechanism by which the daily storage of lipids in lipid droplets, and cellular lipid homeostasis itself, is coordinated. Further, targeted overexpression of TFAP2C induced the equivalent phenotype of steatosis in mouse liver (*Holl et al., 2011*), underscoring the role of TFAP2 proteins in the regulation of neutral lipid metabolism.

The Wnt pathway is also not the first developmental signalling pathway found to exert key regulatory functions in directing energy metabolism. The FOXA family of transcription factors, fundamental for early embryogenesis, play core roles in directing glucose metabolism (*Friedman and Kaestner, 2006*), while SOX17 has been shown to have additional, non-developmental functions regulating lipid metabolism in the liver of adult animals (*Rommelaere et al., 2014*). Given the overlapping requirement for a molecular queue to synchronize energy storage during both embryo development and daily metabolic activity, it is not surprising that some of these systems have evolved to serve dual roles in both biological contexts. These findings support the view that both Wnt signaling, and the TFAP2 family of transcription factors have important (and possibly linked) roles in development and circadian lipid metabolism of the cell.

## Materials and methods

**Key resources table**

| Reagent type (species) or resource | Designation | Source or reference | Identifiers | Additional information |
|---|---|---|---|---|

*Continued on next page*

Continued

| Reagent type (species) or resource | Designation | Source or reference | Identifiers | Additional information |
|---|---|---|---|---|
| Cell line (*Mus musculus*) | L Cells | American Type Culture Collection | Cat #: CRL-2648; RRID:CVCL_4536 | PMID:14056989 |
| Cell line (*Mus musculus*) | L Wnt3A Cells | American Type Culture Collection | Cat #: CRL-2647; RRID:CVCL_0635 | PMID:12717451 |
| Cell line (*Homo sapiens*) | HeLa-MZ | other | | Clone of HeLa (American Type Culture Collection Cat#: CCL-2) provided by Prof. Lucas Pelkmans (University of Zurich) |
| Cell line (*Homo sapiens*) | CRISPR DDIT3 | this paper | | |
| Cell line (*Homo sapiens*) | CRISPR TFAP2A | this paper | | |
| Transfected construct (*Homo sapiens*) | GSK3B | Addgene | Cat #: 49491 | |
| Transfected construct (*Homo sapiens*) | GSK3BS9A | Addgene | Cat #: 49492 | |
| Transfected construct (*Homo sapiens*) | Wnt Project plasmid library | Addgene | Kit # 1000000022 | |
| Transfected construct (*Homo sapiens*) | pCMV-SREBP-1a(460) | American Type Culture Collection | Cat #: 99637 | PMID:9062341 |
| Transfected construct (*Homo sapiens*) | pCMV-SREBP-1c(436) | American Type Culture Collection | Cat #: 99636 | PMID:9062341 |
| Transfected construct (*Homo sapiens*) | pCMV-SREBP-2(468) | American Type Culture Collection | Cat #: 63452 | PMID:9062341 |
| Transfected construct (*Homo sapiens*) | DDIT3-mCherry | this paper | | |
| Transfected construct (*Homo sapiens*) | mCherry-TFAP2A | this paper | | |
| Transfected construct (*Homo sapiens*) | mCherry-TFAP2B | this paper | | |
| Transfected construct (*Homo sapiens*) | mCherry-TFAP2C | this paper | | |
| Transfected construct (*Homo sapiens*) | V5-TFAP2A | this paper | | |
| Biological sample (*Bos taurus*) | Lipoprotein-depleted serum | PMID:13252080 | | |
| Antibody | Rabbit anti-AP2 alpha; anti-TFAP2A | Abcam | Cat #: ab52222 | |
| Antibody | Mouse anti-CHOP (L63F7); anti-DDIT3 | Cell Signaling | Cat #: 2895 | |
| Antibody | Mouse anti-V5 Tag | ThermoFisher Scientific | Cat #: R960-25 | |

*Continued on next page*

*Continued*

| Reagent type (species) or resource | Designation | Source or reference | Identifiers | Additional information |
|---|---|---|---|---|
| Recombinant DNA reagent | CRISPR Forward: DDIT3 | Microsynth | | CACCGGCACCTATATCTCATCCCC |
| Recombinant DNA reagent | CRISPR Forward: TFAP2A | Microsynth | | CACCGGAGTAAGGATCTTGCGACT |
| Recombinant DNA reagent | CRISPR Reverse: DDIT3 | Microsynth | | AAACGACTGATCCAACTGCAGAGAC |
| Recombinant DNA reagent | CRISPR Reverse: TFAP2A | Microsynth | | AAACAGTCGCAAGATCCTTACTCC |
| Recombinant DNA reagent | Primer Forward: ACSL3 | Microsynth | | TGAGCTCTCTTTGCTTGGTG |
| Recombinant DNA reagent | Primer Forward: ACSL4 | Microsynth | | AAGGACATCCCGAAACACAC |
| Recombinant DNA reagent | Primer Forward: AGPAT2 | Microsynth | | GGCCTAAGGCAAAAGGATGTG |
| Recombinant DNA reagent | Primer Forward: AGPAT3 | Microsynth | | ACCCAAGCTCAGCAAGTCC |
| Recombinant DNA reagent | Primer Forward: CTCF | Microsynth | | GCCAGTCCAACCGGCTTATG |
| Recombinant DNA reagent | Primer Forward: LPCAT2 | Microsynth | | AGGGGAAGTGGTTGCTCAATG |
| Recombinant DNA reagent | Primer Forward: MGLL | Microsynth | | GAACCCAGCTCAGTTCAGG |
| Recombinant DNA reagent | Primer Forward: PLIN3 | Microsynth | | TTTGGCAGAGGTGGCAAAC |
| Recombinant DNA reagent | Primer Forward: PLIN4 | Microsynth | | AACCTGCAGGGAAGGTGTTC |
| Recombinant DNA reagent | Primer Forward: PNPLA2 | Microsynth | | TGGCTTCCCTAACTCAGCTTG |
| Recombinant DNA reagent | Primer Forward: PNPLA3 | Microsynth | | TGTCAAGGAAAACAGAAGGAAGC |
| Recombinant DNA reagent | Primer Reverse: ACSL3 | Microsynth | | TGAAAGGTTGCCTTCCTGAG |
| Recombinant DNA reagent | Primer Reverse: ACSL4 | Microsynth | | TCGCCTCAAGTTGTTGCTC |
| Recombinant DNA reagent | Primer Reverse: AGPAT2 | Microsynth | | CTTCAAATGAATGGGGAACTGC |
| Recombinant DNA reagent | Primer Reverse: AGPAT3 | Microsynth | | GCCCGGTACCTTGTGTGAC |
| Recombinant DNA reagent | Primer Reverse: CTCF | Microsynth | | GGTTCTCCCAAGCAGGAGCA |
| Recombinant DNA reagent | Primer Reverse: LPCAT2 | Microsynth | | TCTATGAACCTCGGTTGCCTTC |
| Recombinant DNA reagent | Primer Reverse: MGLL | Microsynth | | CAGCCACGCACTCCTCTC |
| Recombinant DNA reagent | Primer Reverse: PLIN3 | Microsynth | | GATCCACAGGAAGTTCAAGCTG |
| Recombinant DNA reagent | Primer Reverse: PLIN4 | Microsynth | | TTCCTCCTTCGCTTGCTTC |
| Recombinant DNA reagent | Primer Reverse: PNPLA2 | Microsynth | | TCATCTCTGGACCTAGCTGTTGC |
| Recombinant DNA reagent | Primer Reverse: PNPLA3 | Microsynth | | GCAGCGACTCGAGAGAAAGC |

*Continued on next page*

*Continued*

| Reagent type (species) or resource | Designation | Source or reference | Identifiers | Additional information |
|---|---|---|---|---|
| Recombinant DNA reagent | Primer set: ACSL3 | QIAGEN | Cat #: QT01068333 | |
| Recombinant DNA reagent | Primer set: ACTB | QIAGEN | Cat #: QT01136772 | |
| Recombinant DNA reagent | Primer set: AGPAT2 | QIAGEN | Cat #: QT00104888 | |
| Recombinant DNA reagent | Primer set: AGPAT3 | QIAGEN | Cat #: QT00131481 | |
| Recombinant DNA reagent | Primer set: DDIT3 | QIAGEN | Cat #: QT01749748 | |
| Recombinant DNA reagent | Primer set: DGAT2 | QIAGEN | Cat #: QT00134477 | |
| Recombinant DNA reagent | Primer set: HMGCR | QIAGEN | Cat #: QT00004081 | |
| Recombinant DNA reagent | Primer set: LDLR | QIAGEN | Cat #: QT00045864 | |
| Recombinant DNA reagent | Primer set: MGLL | QIAGEN | Cat #: QT01163428 | |
| Recombinant DNA reagent | Primer set: PLIN4 | QIAGEN | Cat #; QT00112301 | |
| Recombinant DNA reagent | Primer set: PNPLA2 | QIAGEN | Cat #: QT00111846 | |
| Recombinant DNA reagent | Primer set: SOAT1 | QIAGEN | Cat #: QT01046472 | |
| Recombinant DNA reagent | Primer set: SREBPF1 | QIAGEN | Cat #: QT00167055 | |
| Recombinant DNA reagent | Primer set: SREBPF1 | QIAGEN | Cat #: QT00036897 | |
| Recombinant DNA reagent | Primer set: SREBPF2 | QIAGEN | Cat #: QT00255204 | |
| Recombinant DNA reagent | Primer set: SREBPF2 | QIAGEN | Cat #: QT00052052 | |
| Recombinant DNA reagent | Primer set: TFAP2A | QIAGEN | Cat #: QT00085225 | |
| Recombinant DNA reagent | Primer set: TFAP2C | QIAGEN | Cat #: QT00073073 | |
| Sequence-based reagent | siRNA DDIT3 | Dharmacon | Cat #: J-062068 | |
| Sequence-based reagent | siRNA TFAP2A | Dharmacon | Cat #: J-062799 | |
| Sequence-based reagent | siRNA TFAP2C | Dharmacon | Cat #: J-048594 | |
| Sequence-based reagent | siRNA APC | QIAGEN | Cat #: S102757251 | |
| Sequence-based reagent | siRNA DGAT1 | QIAGEN | Cat #: S100978278 | |
| Sequence-based reagent | siRNA DGAT2 | QIAGEN | Cat #: S100978278 | |
| Sequence-based reagent | siRNA GSK3B | QIAGEN | Cat #: S100300335 | |
| Sequence-based reagent | siRNA SOAT1 | QIAGEN | Cat #: S101428924 | |

*Continued on next page*

*Continued*

| Reagent type (species) or resource | Designation | Source or reference | Identifiers | Additional information |
|---|---|---|---|---|
| Commercial assay or kit | Bio-Rad Protein Assay Kit | Bio-Rad Laboratories | Cat #: 500–0006 | |
| Commercial assay or kit | SsoAdvanced SYBR Green Supermix | Bio-Rad Laboratories | Cat #: 1725270 | |
| Commercial assay or kit | Triglyceride Colorimetric Assay Kit | Cayman Chemicals | Cat #: 10010303 | |
| Commercial assay or kit | Wizard SV gel and PCR Clean-up system | Promega | Cat #: A9281 | |
| Commercial assay or kit | RNeasy Mini Kit | QIAGEN | Cat #: 74104 | |
| Commercial assay or kit | RNeasy Mini Kit | QIAGEN | Cat #: 74104 | |
| Commercial assay or kit | Amplex Red Cholesterol Assay Kit | ThermoFisher Scientific | Cat #: A12216 | |
| Commercial assay or kit | SuperScript VILO cDNA Synthesis Kit | ThermoFisher Scientific | Cat #: 11754050 | |
| Chemical compound, drug | Wnt Pathway Library | Enzo Life Sciences | Cat #: BML-2838 | |
| Chemical compound, drug | BODIPY 493/503 | ThermoFisher Scientific | Cat #: D3922 | |
| Chemical compound, drug | Hoechst 33342 | ThermoFisher Scientific | Cat #: H3570 | |
| Chemical compound, drug | Lipofectamine 3000 | ThermoFisher Scientific | Cat #: L3000015 | |
| Chemical compound, drug | Lipofectamine LTX | ThermoFisher Scientific | Cat #: A12621 | |
| Chemical compound, drug | Lipofectamine RNAiMax | ThermoFisher Scientific | Cat #: 13778100 | |
| Chemical compound, drug | A-922500 | Tocris Bioscience | Cat #: 3587 | PMID:18183944 |
| Chemical compound, drug | PF-429242 | Tocris Bioscience | Cat #: 3354 | PMID:17583500 |
| Chemical compound, drug | Torin-2 | Tocris Bioscience | Cat #: 4248 | PMID:21322566 |

## Cells, media, reagents and antibodies

HeLa-MZ cells, a line of HeLa cells selected to be amiable to imaging, were provided by Prof. Lucas Pelkmans (University of Zurich). HeLa cells are not on the list of commonly misidentified cell lines maintained by the International Cell Line Authentication Committee.

Our HeLa-MZ cells were authenticated by Microsynth (Balgach, Switzerland), which revealed 100% identity to the DNA profile of the cell line HeLa (ATCC: CCL-2) and 100% identity over all 15 autosomal STRs to the Microsynth's reference DNA profile of HeLa. L cells (ATCC: CRL-2648) and L Wnt3A cells (ATCC: CRL-2647) were generously provided by Prof. Gisou van der Goot (École Polytechnique Fédérale de Lausanne; EPFL) and cultured as per ATCC recommendations. Cells are mycoplasma negative as tested by GATC Biotech (Konstanz, Germany). Wnt3A-conditioned, and control-conditioned media was prepared from these cells by pooling two subsequent collections of 24 hr each from confluent cells. Reagents were sourced as follows: Hoechst 33342 and BODIPY 493/503 from Molecular Probes (Eugene, OR); The DGAT1 inhibitor A-922500 and the Membrane Bound

Transcription Factor Peptidase, Site 1 (S1P/SREBF) inhibitor PF-429242, and the mTOR inhibitor Torin-2 were from Tocris Bioscience (Zug, Switzerland); lipoprotein-depleted serum (LPDS) was prepared as previously described (*Havel et al., 1955*); anti-DDIT3 antibodies were from Cell Signaling (L63F7; Leiden, The Netherlands); anti-TFAP2A antibodies were from Abcam (ab52222; Cambridge, UK); fluorescently labeled secondary antibodies from Jackson ImmunoResearch Laboratories (West Grove, PA); 96-well Falcon imaging plates (#353219) were from Corning (Corning, NY); oligonucleotides and small interfering RNA (siRNA) from Dharmacon (SmartPool; Lafayette, CO) or QIAGEN (Venlo, The Netherlands) and the siRNAs used in this work were: GSK3B (S100300335); DGAT1 (S100978278); DGAT2 (S100978278); SOAT1 (S101428924); TFAP2A (J-062799); TFAP2C (J-048594); DDIT3 (J-062068); APC (S102757251).

Other chemicals and reagents were obtained from Sigma-Aldrich (St. Louis, MO).

Transfections of cDNA and siRNAs were performed using Lipofectamine LTX and Lipofectamine RNAiMax (Invitrogen; Basel, Switzerland) respectively using the supplier's instructions.

Plasmids encoding GSK3B and GSK3B$^{S9A}$, and were from Addgene (Cambridge, MA) and the SREBF truncations from ATCC; TFAP2A, TFAP2B, and DDIT3 coding sequences were obtained from the Gene Expression Core Facility at the EPFL, and TFAP2C from DNASU (Arizona State University) and cloned into appropriate mammalian expression vectors using Gateway Cloning (Thermo Fisher Scientific). Knock-out cell lines of TFAP2A and DDIT3 were obtained by clonal isolation after transfection of a pX330 Cas9 plasmid (*Cong et al., 2013*) with appropriate guide sequences (see Supplementary Methods).

## Lipid droplet quantitation

L Cells were seeded (6 000 cells/well) in imaging plates the day before addition of control-conditioned, or Wnt3a-conditioned media for 24 hr, before fixation with 3% paraformaldehyde for 20 min. Where necessary, cells were treated with siRNAs and Lipofectamine RNAiMax Reagent (Thermo Fisher) per the manufacturer's instructions for 48 hr before seeding into imaging plates. Lipid droplets and nuclei were labelled with 1 µg/mL BODIPY 493/503 (Invitrogen; D3922) and 2 µg/mL Hoechst 33342 (Invitrogen; H3570) for 30 min, wash with PBS, and the plate sealed and imaged with ImageXpress Micro XLS (Molecular Devices, Sunnyvale, CA) automated microscope using a 60X air objective. Images were segmented using CellProfiler (*Carpenter et al., 2006*) to identify and quantify nuclei, cells, and lipid droplets.

## Wnt ligand screen

L Cells cells were seeded into image plates (6000 cells/well) the morning before transfection with Lipofectamine 3000 (Invitrogen) as per the manufacturer's instructions with a subset of untagged Wnts from the open source Wnt Project plasmid library (*Najdi et al., 2012*). Cells were fixed with 3% PFA after 48 hr, and lipid droplets quantified as above. Data were normalized to the number of lipid droplets in the empty vector condition. The circular phylogenetic tree was constructed using human Wnt sequences and the Lasergene (v12.1; DNAStar, Madison WI) bioinformatics software.

## Wnt compound screen

L Cells were seeded into imaging plates (6000 cells/well) the day before addition of the Wnt Pathway Library (BML-2838; Enzo Life Sciences, Farmingdale, NY) at either 1 µM or 10 µM, with either control-conditioned or Wnt3a-conditioned media for 24 hr, before fixation and lipid droplet quantitation as above. Data were normalized by z-score and the average of quadruplicates.

## mRNA determination

Total-RNA extraction was performed using RNeasy Mini Kit from Qiagen (74104) from L Cells or HeLa-MZ according to manufacturer's recommendation. cDNA synthesis was carried out using SuperScript VILO cDNA Synthesis Kit (Life Technologies AG; Basel, Switzerland) from 250 ng of total RNA. mRNA expression was evaluated using SsoAdvanced SYBR Green Supermix (Bio-Rad Laboratories, Hercules, CA) with 10 ng of cDNA with specific primers of interest on a CFX Connect real-time PCR Detection System(Bio-Rad). Relative amounts of mRNA were calculated by comparative CT analysis with 18S ribosomal RNA used as internal control. All primers are QuantiTect primer from Qiagen (Hilden, Germany) and were: SOAT1 (QT01046472), DGAT2 (QT00134477), ACSL3

(QT01068333), AGPAT2 (QT00104888), AGPAT3 (QT00131481), MGLL (QT01163428), PLIN4 (QT00112301), PNPLA2 (QT00111846), SREBPF1 (QT00167055), SREBPF1 (QT00036897), SREBPF2 (QT00255204), SREBPF2 (QT00052052), DDIT3 (QT01749748), TFAP2A (QT00085225), TFAP2C (QT00073073), ACTB (QT01136772), LDLR (QT00045864), HMGCR (QT00004081).

## Lipid determinations

Cholesterol esters and triglycerides amounts were determined using the Amplex Red Cholesterol Assay Kit (A12216; Thermo Fisher Scientific) and the Triglyceride Colorimetric Assay Kit (10010303; Cayman Chemicals), respectively. Briefly, cells were cultured as needed and scraped directly into lysis buffer (150 mM NaCl, 20 mM HEPES pH 7.4, 1% Triton X-100). Protein concentrations were determined for normalization using Bio-Rad Protein Assay Kit (500–0006) and lipid determinations were made as per the manufacturer's instructions. Cholesterol determinations were made twice, with and without cholesterol esterase, in order to infer the amount of cholesteryl esters.

## RNAseq

HeLa-MZ cells were treated in triplicate with control-conditioned, or Wnt3a-conditioned media for 2 hr or 6 hr before cells were collected and RNA isolated. Purification of total RNA was done with the RNeasy Mini Kit from Qiagen according to the manufacturer's instructions. The concentration, purity, and integrity of the RNA were measured with the Picodrop Microliter UV-Vis Spectrophotometer (Picodrop), and the Agilent 2100 Bioanalyzer (Agilent Technologies), together with the RNA 6000 Series II Nano Kit (Agilent) according to the manufacturer's instructions. Purified RNAs were sequenced with a HiSeq 4000 (Illumina) at the iGE3 Genomics Platform of the University of Geneva (https://ige3.genomics.unige.ch).

Sequencing reads were mapped to the hg38 genome using bowtie2 in local alignment mode. Then reads were attributed to known exons as defined by the ensembl annotation, and transcript-level read counts were inferred as described in *David et al. (2014)*. Differential expression was then evaluated by LIMMA (*Law et al., 2014*) using the log of rpkm values.

The fold induction was determined as a ratio of mRNA amounts of Wnt3a to control. Genes with message levels increase more than 1.5-fold, or decreased less than 0.8-fold were collected and tested for pathway enrichment using DAVID bioinformatics resources (v6.8) (*Huang et al., 2009*) and the resulting data compiled and plotted using Cytoscape (*Shannon et al., 2003*) and previously described Matlab scripts (*Mercer et al., 2012*).

## Wnt siRNA screen analysis

Data from the genome-wide siRNA screen (*Scott et al., 2015*) was used to produce the subset of annotated transcription factors (GO:0003700) cellular cholesterol levels after gene silencing provided by AmiGO 2 (version 2.4.26) (*Carbon et al., 2009*). Data were filtered for genes that increased or decreased total cellular cholesterol a z-score of 1.5 or greater.

## Transcription factor enrichment

Transcript data from cells treated with Wnt3a (E-MTAB-2872 [*Scott et al., 2015*]), or fatty acids (GSE21023 [*Lockridge et al., 2008*], GSE22693 [*Xu et al., 2015*], GSE42220 [(*Shaw et al., 2013*]) were collected and mRNAs that significantly changed in response to treatment were compiled and analyzed for transcription factor enrichment using GeneGo (MetaCore) bioinformatics software (Thomson Reuters). Data from both conditions was compared and ordered by a sum of ranking.

## Promoter analysis

Promoter sequences from the current (GRCh38) human genome were collected using Regulatory Sequence Analysis Tools (RSAT) software (*Medina-Rivera et al., 2015*) to collect the non-overlapping upstream promoter sequences (<3 000 bp) of genes of interest which were tested for enriched sequences using either the RSAT's oligo-analysis or Analysis of Motif Enrichment (AME) module of MEME Suite (v.4.12.0) (*McLeay and Bailey, 2010*). Genes with no non-overlapping upstream region were discarded from the analysis. Identification and quantitation of consensus binding sites was done with either RSAT's matrix-scan or MEME Suite's Motif Alignment and Search Tool (MAST) module using a cut-off of <0.0001 to define a consensus site. Consensus sites from JASPAR

(*Mathelier et al., 2016*) were: DDIT3:CEBPA (MA0019.1), TFAP2A (MA0003.2, MA0810.1), TFAP2C (MA0524.1, MA0524.2).

## ChIP-qPCR

Hela-MZ cells were transfected as above with empty vector or pcDna6.2-V5-TFAP2A. After 24 hr Hela medium was replaced with either: fresh medium (mock), control or Wnt3a-conditional medium. At 48 hr from transfection cell were incubated with PFA 1% for 8 min; after quenching the reaction with glycine 0.125M, cells were resuspended sequentially in the following lysis buffers: lysis buffer I (50 mM Hepes-KOH, 140 mM NaCl, 1 mM EDTA, 10% glycerol, 0.5% of NP-40, 0.25% Triton X-100; adjust final pH to 7.5), lysis buffer II (10 mM Tris-HCl pH: 8.0, 200 mM NaCl, 1 mM EDTA, 0.5 mM EGTA; adjust final pH to 8.0), lysis buffer III (10 mM Tris-HCl pH: 8.0, 100 mM NaCl, 1 mM EDTA, 0.5 mM EGTA, 0.1% Na-deoxycholate, 0.5% N-lauroylsarcosine; adjust final pH to 8.0) all containing protease inhibitors. Chromatin is sonicated using a Bioruptor Sonicator (Diagenode Inc.) to generate 200- to 1000 bp DNA fragments (30 pulses of 30 s at high power- total time of 30 min, 30 s ON, 30 s OFF). After microcentrifugation, the supernatant is diluted in buffer III in order to have 2 mg of protein in 1 ml (1:10 of each sample has been removed and use as input for the relative sample). Chromatin was incubated with 5 µg of antibody (V5 Tag Monoclonal Antibody; R960-25 Thermo Fisher Scientific) on a rotator for 14–16 hr at 4°C and 4 h hours more after adding 50 µl of magnetic beads (Dynabeads Protein G fron Invitrogen). After the reverse cross-linking and elution immunoprecipitated chromatin was purified by columns (Promega Wizard SV gel and PCR Clean-up system; A9281). Quantitative PCR was performed using the SsoAdvanced SYBR Green Supermix, and an CFX Connect real-time PCR Detection System (Bio-Rad Laboratories, Hercules, CA) according to manufacturer's instructions. Primers used are listed below. Dissociation curves after amplification showed that all primer pairs generated single products. The amount of PCR product amplified was calculated as percent of the input. A genomic region of an intron in Myc gene (CTCF, CCCTC-Binding Factor) was used as negative control.

## List of primers for ChIP-qPCR

MGLL
Forward: GAACCCAGCTCAGTTCAGG
  Reverse: CAGCCACGCACTCCTCTC

PLIN4
Forward: AACCTGCAGGGAAGGTGTTC
  Reverse: TTCCTCCTTCGCTTGCTTC

ACSL3
Forward: TGAGCTCTCTTTGCTTGGTG
  Reverse: TGAAAGGTTGCCTTCCTGAG

AGPAT3
Forward: ACCCAAGCTCAGCAAGTCC
  Reverse: GCCCGGTACCTTGTGTGAC

ACSL4
Forward: AAGGACATCCCGAAACACAC
  Reverse: TCGCCTCAAGTTGTTGCTC

AGPAT2
Forward: GGCCTAAGGCAAAAGGATGTG
  Reverse: CTTCAAATGAATGGGGAACTGC

PNPLA3
Forward: TGTCAAGGAAAACAGAAGGAAGC

Reverse: GCAGCGACTCGAGAGAAAGC

## PNPLA2
Forward: TGGCTTCCCTAACTCAGCTTG
Reverse: TCATCTCTGGACCTAGCTGTTGC

## LPCAT2
Forward: AGGGGAAGTGGTTGCTCAATG
Reverse: TCTATGAACCTCGGTTGCCTTC

## PLIN3
Forward: TTTGGCAGAGGTGGCAAAC
Reverse: GATCCACAGGAAGTTCAAGCTG

## CTCF site inside the gene Myc:
Foward: GCCAGTCCAACCGGCTTATG
Reverse: GGTTCTCCCAAGCAGGAGCA

## Construction of gene edited knock out cell lines

The guide sequences used to construct specific CRISPR/Cas9 vectors were determined using the CRISPR Design Tool(*Ran et al., 2013*) and were:

## TFAP2A
Fwd: CACCGGAGTAAGGATCTTGCGACT
Rev: AAACAGTCGCAAGATCCTTACTCC

## DDIT3
Fwd: CACCGGCACCTATATCTCATCCCC
Rev: AAACGACTGATCCAACTGCAGAGAC

These sequences were used to create insert the target sequence into the pX330 vector using Golden Gate Assembly (New England Biolabs) and transfected into cells as described in the Methods. Knock-out clones were isolated by serial dilution and confirmed by western blotting and activity assays.

# Acknowledgements

We are grateful to Christian Iseli, Nicolas Guex, and Ioannis Xenarios from Vital-IT and the Swiss Institute of Bioinformatics for indispensable guidance on the interpretation of the transcriptomics data and critical reading of the manuscript. We would like to also thank Dimitri Moreau and ACCESS Geneva for invaluable assistance with high-content screening. Support was from the Swiss National Science Foundation, the NCCR in Chemical Biology and LipidX from the Swiss SystemsX.ch Initiative, evaluated by the Swiss National Science Foundation (to JG). CCS has been supported by fellowships from the Human Frontier Science Program and the Canadian Institutes of Health Research.

# Additional information

## Funding

| Funder | Grant reference number | Author |
| --- | --- | --- |
| NCCR Chemical Biology | 51NF40_125781 | Jean Gruenberg |
| Swiss SystemsX.ch Initiative - LipidX | 51RT-0_145726 | Jean Gruenberg |
| Human Frontier Science Program | Fellowship | Cameron C Scott |

| Canadian Institutes of Health Research | Fellowship | Cameron C Scott |
| Schweizerischer Nationalfonds zur Förderung der Wissenschaftlichen Forschung | 31003A_159479 | Jean Gruenberg |

The funders had no role in study design, data collection and interpretation, or the decision to submit the work for publication.

### Author contributions

Cameron C Scott, Conceptualization, Data curation, Formal analysis, Validation, Investigation, Visualization, Methodology, Writing—original draft, Writing—review and editing; Stefania Vossio, Investigation, Visualization; Jacques Rougemont, Formal analysis; Jean Gruenberg, Conceptualization, Resources, Supervision, Funding acquisition, Validation, Writing—original draft, Project administration, Writing—review and editing

### Author ORCIDs

Cameron C Scott ⓘ http://orcid.org/0000-0001-5026-3413
Jacques Rougemont ⓘ http://orcid.org/0000-0002-0955-606X
Jean Gruenberg ⓘ http://orcid.org/0000-0002-0300-4862

### Decision letter and Author response

Decision letter https://doi.org/10.7554/eLife.36330.022
Author response https://doi.org/10.7554/eLife.36330.023

# Additional files

### Supplementary files

• Transparent reporting form
DOI: https://doi.org/10.7554/eLife.36330.018

### Data availability

All data generated or analysed during this study are included in the manuscript and supporting file. RNAseq data have been deposited under accession number E-MTAB-6623.

The following dataset was generated:

| Author(s) | Year | Dataset title | Dataset URL | Database, license, and accessibility information |
|---|---|---|---|---|
| Scott CC, Gruenberg J | 2018 | HeLa cells treated with Wnt3a | https://www.ebi.ac.uk/arrayexpress/experiments/E-MTAB-6623 | Publicly available at the Electron Microscopy Data Bank (accession no: E-MTAB-6623) |

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
