## [Decision Letter]

Thank you for submitting your article "TFAP2 transcription factors are regulators of lipid droplet biogenesis" for consideration by *eLife*. Your article has been reviewed by Vivek Malhotra as the Senior Editor, Tobias Walther as a guest Reviewing Editor, and three reviewers. The reviewers have opted to remain anonymous.

After extensive consultation amongst the reviewers, we have drafted this decision to help you prepare a revised submission.

All the reviewers found the work interesting and a significant advance in our understanding of the transcriptional control of the Wnt pathway regulating neutral lipid storage in Lipid Droplets. The reviewers, however, raised a number of concerns, of which the most essential follow.

- Key experiments need measurement of (neutral) lipids. Currently only BODIPY images are shown. In addition, these images need to be quantified.

- Data on transcription factor binding (ChIP-seq or at least CHIP-qPCR) need to test binding on the proposed sites.

- Further testing the function of TFAP2 in WNT mediated LD formation using knock out cells. This should ideally include some clarification on the role/order of TFAP2A and DDIT3/CHOP.

---

## [Author Response]

We are very happy that the reviewers found our work interesting, and we sincerely thank them for their helpful and constructive comments. We have addressed all queries and issues that were raised: you will find below our detailed reply to each comment.

- Key experiments need measurement of (neutral) lipids. Currently only BODIPY images are shown. In addition, these images need to be quantified.

In all light microscopy experiments in the manuscript, lipid droplets (LDs) stained with BODIPY have been quantified by automated microscopy using the ImageXpress Micro XLS microscope from Molecular Devices. We also wish to add that BODIPY does not stain other organelles — including endosomes/lysosomes in Niemann-Pick type C cells, which accumulate cholesterol in endosomes/lysosomes. In this revised version, as an alternative LD marker, we used the LD protein perilipin A (PLIN1a) tagged with the GFP, expressed in cells treated or not with Wnt. We show that the LD association of PLIN1a-GFP recapitulates the BODIPY data (new Figure 3—figure supplement 1). In addition, we have also quantified triglycerides (TAG) and cholesteryl esters in cells treated or not with Wnt, and we find that accumulation of these characteristic lipid droplet molecules in Wnt-treated cells confirms our findings with BODIPY and PLIN1a-GFP (new Figure 3—figure supplement 1).

- Data on transcription factor binding (ChIP-seq or at least CHIP-qPCR) need to test binding on the proposed sites.

We thank the reviewers and editor for this helpful suggestion. As requested, we have carried out CHIP-qPCR experiments, and tested the ability of TFAP2A to directly bind to the promoter region of known lipid droplet, and lipid metabolic genes containing a predicted TFAP2 consensus site, including the enzymes ACSL3, ACSL4, AGPAT2, AGPAT3, LPCAT2, and MGLL, and the lipid droplet resident proteins PLIN3, PLIN4, PNPLA2, and PNPLA3. We now show that the TFAP2A protein has a broad capacity to bind the promoter region of all the genes we tested, supporting the notion that expression is controlled by TFAP2A (new Figure 2C).

- Further testing the function of TFAP2 in WNT mediated LD formation using knock out cells. This should ideally include some clarification on the role/order of TFAP2A and DDIT3/CHOP.

As requested we have further tested the role of TFAP2A and DDIT3 in the CRISPR/Cas9 KO cells. The ability of Wnt to induce LD formation could be rescued in both TFAP2A and DDIT3 KO cell lines, upon re-expression of TFAP2 and DDIT3, respectively, demonstrating that the inhibition in both cell lines was not due to some off-target or indirect effects of the treatments. In an attempt to order/clarify the role of these transcription factors, we also tested whether

TFAP2A overexpression restored Wnt-dependent LD formation in DDIT3 KO cells and vice versa. We find that TFAP2A did not rescue DDIT3 KO, while DDIT3 was able to rescue TFAP2A KO. Based on the known repressor role of DDIT3 in SREBP expression, it is attractive to speculate that DDIT3 expression responds to the presence of lipid droplets. DDIT3 therefore, in addition to a direct inducer of lipid storage, might act as TFAP2A repressor repressor — in line with the known repressor function of DDIT3 in SREBP expression. This issue is now clarified in the text.